## PERSPECTIVE

# I(*nsp1*)ecting SARS-CoV-2–ribosome interactions

Matthieu Simeoni[1,2], Théo Cavinato[1,2], Daniel Rodriguez[1,2] & David Gatfield [1✉]

While SARS-CoV-2 is causing modern human history's most serious health crisis and upending our way of life, clinical and basic research on the virus is advancing rapidly, leading to fascinating discoveries. Two studies have revealed how the viral virulence factor, non-structural protein 1 (Nsp1), binds human ribosomes to inhibit host cell translation. Here, we examine the main conclusions on the molecular activity of Nsp1 and its role in suppressing innate immune responses. We discuss different scenarios potentially explaining how the viral RNA can bypass its own translation blockage and speculate on the suitability of Nsp1 as a therapeutic target.

Severe acute respiratory syndrome coronavirus 2 (SARS-CoV-2) causes the disease COVID-19 that has led to one of the most serious health crises in modern history[1]. First identified in Wuhan, China, the virus subsequently spread around the world and was declared a pandemic in March 2020[2]. At the time of writing (December 2020), SARS-CoV-2 had worldwide killed more than 1.5 million people and infected almost 70 million according to the World Health Organization[3]. Shortly after China reported its first confirmed cases of infection, the causative agent of COVID-19 was identified as a member of the Sarbecovirus subgenus of the genus *Betacoronavirus*[4,5], which also includes two already known causative agents of epidemics: severe acute respiratory syndrome coronavirus (SARS-CoV or SARS-CoV-1) and Middle East respiratory syndrome coronavirus (MERS-CoV)[6]. Although SARS-CoV-2 shares part of its genome with SARS-CoV-1 and MERS-CoV (approximately 80% and 50%, respectively)[5,7], it has a higher rate of spread and its symptoms develop after a longer incubation period, making it a major threat to global health.

SARS-CoV-2 is an enveloped positive-stranded RNA virus[5]. Its 30 kb genome comprises a 5′-cap and 5′ untranslated region (5′ UTR), followed by ten individual protein-coding open reading frames (ORFs), and terminates with a 3′ UTR that is polyadenylated (Fig. 1a). The 3′ portion of the genome encodes several typical viral structural proteins, such as spike, envelope, membrane and nucleocapsid proteins, whereas in the genome's 5′ portion two large overlapping ORFs of gene 1 encode the ORF1a/b polyprotein, from which several nonstructural proteins (Nsps) arise through proteolytic cleavage. Among the 16 Nsps (Nsp1–16), Nsp1 is encoded at the very 5′ end of ORF1a (Fig. 1a) and is the first coronaviral protein produced in infected cells[8]. Previous work on SARS-CoV-1 reported several activities for Nsp1: it can suppress host translation by interacting with the ribosomal 40S subunit and inhibiting 80S formation[9,10], and it can induce mRNA cleavage and decay[11,12], leading to an inhibition of cell-intrinsic innate immune responses[13,14]. Of note, the mechanisms by which Nsp1 proteins operate may vary across beta-CoVs[15]: for instance, it has been reported that MERS-CoV Nsp1 does not stably bind the ribosomal 40S subunit and—in line with its intracellular distribution that is both cytoplasmic and nuclear—that it possesses an RNA degradation activity that differs from that of the exclusively cytoplasmic SARS-CoV-1 Nsp1[16]. Whether and how SARS-CoV-2 Nsp1 can inhibit translation has remained poorly understood until recently, with two studies by Schubert et al.[17] and Thoms et al.[18] now providing insights into how Nsp1 binds to the 40S subunit of the ribosome and blocks the mRNA entry channel. Using cryo-electron microscopy, the two studies highlight areas of interaction between Nsp1 and the ribosome and show that the 5′ UTR of the viral transcript

[1]Center for Integrative Genomics, University of Lausanne, Lausanne, Switzerland. [2]These authors contributed equally: Matthieu Simeoni, Théo Cavinato, Daniel Rodriguez. ✉email: david.gatfield@unil.ch

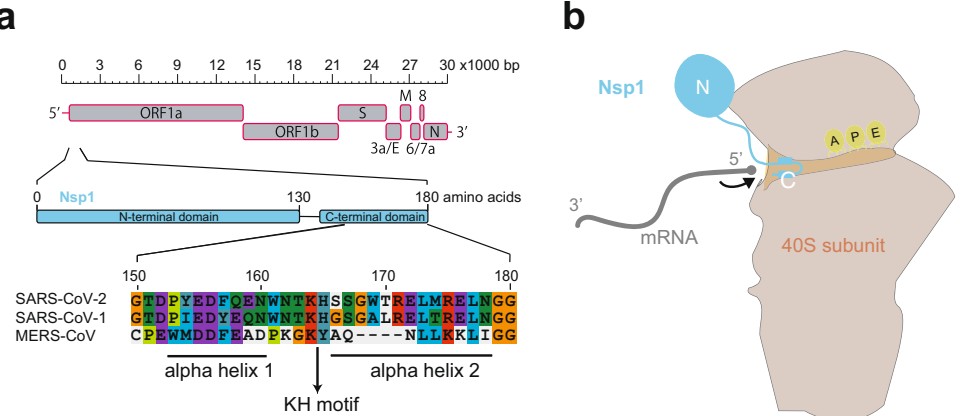

**Fig. 1 Nsp1 interaction with the ribosome. a** Schematic of SARS-CoV-2 genome organisation with the whole genome depicted at the top, Nsp1 coding sequence in the middle, and a sequence alignment of Nsp1 C-terminal domain of SARS-CoV-2, SARS-CoV-1, and MERS-CoV in the lower part of the panel. The two alpha helices and the KH motif are marked by bars and an arrow, respectively. Colour coding of amino acids corresponds to default settings of the ClustalX alignment tool. **b** Cartoon depicting the interaction between Nsp1 and the 40S ribosomal subunit, as revealed by the structural data. The C-terminal helices anchor Nsp1 in the mRNA entry channel, thereby blocking access for host transcripts (schematically represented in grey). The globular N-terminus is not sufficiently resolved in the structures to be able to assign a clear position and function.

enhances its translation. Notably, the inhibition by Nsp1 has direct effects on the host immune response, in line with previous work[9].

In this review, we describe how the recent structural work[17,18] has improved our understanding of SARS-CoV-2 Nsp1-mediated translation inhibition. We also discuss which mechanisms may be responsible to sustain viral protein translation even under conditions when Nsp1 inhibits the ribosome. Finally, because Nsp1 is essential for efficient SARS-CoV-2 replication, understanding the molecular mechanisms that underlie its activity may be relevant for the development of effective therapeutic treatments and vaccines. We highlight how Nsp1 inhibition would likely impact host immune responses and inhibit viral replication.

**Nsp1 blocks the mRNA entry channel**. Nsp1 from SARS-CoV-2 has 84% amino acid sequence identity with its SARS-CoV-1 ortholog. Such high conservation suggests common biological properties and functions. For SARS-CoV-1, Nsp1 can lead to an almost complete halt in host translation (and, thus, antiviral defence mechanisms that depend on de novo gene expression), and the protein interacts with the human 40S ribosomal subunit with the help of a Lys164-His165 (K164, H165) dipeptide motif[14]. These residues are conserved in SARS-CoV-2 (Fig. 1a), arguing for functional orthology. How, precisely, does SARS-CoV-2 Nsp1 bind to the ribosome and what is the mechanism underlying translational inhibition? To address these questions, Schubert et al.[17] and Thoms et al.[18] followed similar strategies: first, they used cryo-electron microscopy to determine the structure of Nsp1 bound to host ribosomal complexes. Second, they designed cellular and biochemical experiments to investigate the main hypotheses on how Nsp1 affects translation. While the central conclusions from both studies are overlapping and complementary, the actual Nsp1–ribosome complexes that they report on are at first sight surprisingly diverse. The main reason likely lies in different methodological approaches. Briefly, Schubert et al. incubated Nsp1 that was recombinantly produced in bacteria, with human embryonic kidney (HEK) 293E cell extracts, and purified the resulting Nsp1-ribosomal complexes on sucrose gradients[17]. The structures of two main complexes were solved at atomic resolution, the first corresponding to Nsp1 with a 40S ribosomal subunit and the second together with an 80S ribosome.

The 40S subunit-containing structure showed all features of a 43s pre-initiation complex (PIC) (i.e. it contained the eIF3 core, eIF1 and initiator tRNA-loaded eIF2 proteins) with Nsp1 occupying the mRNA entrance channel. The 80S structure corresponded to a translationally inactive ribosome with an exit site tRNA, but lacking mRNA; again, the mRNA entrance channel was blocked by Nsp1. The main strategy pursued by Thoms et al.[18] was based on expressing tagged Nsp1 in HEK293T cells, followed by Nsp1 affinity purification to isolate native complexes from the cell lysates. The structures of nine distinct Nsp1-containing 40S and 80S complexes were solved. Among the five different 40S complexes, two were in a PIC state, similar to that reported by Schubert et al.[17], whereas three others did not correspond to initiation intermediates. Briefly, two of them contained a ribosomal biogenesis factor, TSR1, indicating a "pre-40S state", while the third was a simple Nsp1–40S association. Of the four distinct Nsp1–80S complexes, two contained an additional protein (CCDC124) that occupied the aminoacyl site (A-site), possibly indicating a ribosome recovery/recycling state. In the two other 80S complexes, a protein that has previously been implicated in pre-rRNA processing and antiviral responses, termed LYAR, occupied the A-site. It is unclear whether these rather exotic complexes and conformations had been induced by the presence of Nsp1 or whether Nsp1 had trapped natural intermediates that thus became purifiable. Moreover, it is unknown what relevance these complexes have in SARS-CoV-2-infected cells.

In all the above complexes, Nsp1 obstructed the mRNA entry channel, consistent with translational inactivity. How, precisely, is mRNA entry blocked by Nsp1? The two studies uncovered the molecular basis of a tight interaction that relies on the C-terminal region of Nsp1, which folds into two helices that insert into the mRNA entrance channel (Fig. 1b). The first C-terminal helix (residues 153–160) makes hydrophobic interactions with 40S ribosomal proteins uS3 and uS5, and the second C-terminal helix (residues 166–178) interacts with ribosomal protein eS30 and helix h18 of the 18S rRNA. In between the two helices, the conserved KH dipeptide (K164 and H165) forms critical interactions with h18 that are based on H165 stacking between two uridines of 18S rRNA (U607 and U630), and electrostatic interactions between K164 and the phosphate backbone of rRNA bases G625 and U630.

In summary, the cryo-EM structures give detailed insights into how Nsp1 uses its C-terminus to cling onto the mRNA entry channel, thus precluding transcript recruitment. Of note, this mechanism may be particular to SARS-CoV-2 and its closest relatives, given that the Nsp1 C-terminus is shorter and less conserved in more distantly related viruses, including MERS-CoV (Fig. 1a). Two obvious questions arise from the structural data. First: what is the function of the protein's N-terminal domain? The cryo-EM data of both studies indicate that the N-terminus adopts a globular shape, flexibly connected to the C-terminus—yet its precise structure remains undefined. When the Nsp1 N-terminus is replaced by an unrelated protein sequence, this fusion still inhibits translation in in vitro assays, indicating that this part of Nsp1 is not required for translation inhibition per se[17]. The N-terminus may thus act in other processes, possibly in analogy to Nsp1 from SARS-CoV-1 that can regulate mRNA stability and suppress host immune functions[13,14]. The second intriguing question is: how does the virus ensure translation of its own RNA? We will discuss various hypotheses in the next section.

**Viral gene expression needs to bypass global translation inhibition.** If Nsp1 binds with high affinity to the ribosome to inhibit translation in a potentially global fashion, an obvious paradox arises: how can the virus produce the proteins necessary for its own replication? The above studies[17,18], together with other recent publications, have given rise to several hypotheses on how viral protein translation may be achieved (Fig. 2).

**The viral 5′ UTR overrides the translation block.** Schubert et al. demonstrate that the highly structured viral 5′ UTR is likely critical to overcome the Nsp1-mediated translation block[17]. In in vitro translation assays, fivefold more protein was produced from a luciferase reporter RNA carrying the viral 5′ UTR as compared to an identical amount of reporter RNA without the viral 5′ UTR. Nevertheless, Nsp1 inhibited the translation of both reporters in a similar, dose-dependent fashion. This finding suggests that at Nsp1 expression levels that do not shut down translation completely, the viral transcript will have a kinetic advantage over cellular transcripts to be recruited for translation. Two recent studies[19,20] go further in characterising the mechanisms involved in lifting the translation block so that viral protein biogenesis can proceed. Analogous to previous observations that had been made using SARS-CoV-1 Nsp1[21], Shi et al., in their non-peer-reviewed publication available as a preprint,

demonstrate that the N-terminal domain of SARS-CoV-2 Nsp1 interacts with the viral 5′ UTR[19]. Moreover, when the physical distance between the Nsp1 C-terminus (that anchors the protein on the 40S subunit, as described above) and the N-terminus (that interacts with the 5′ UTR[19]) is increased through a linker, the viral 5′ UTR-containing RNA loses the ability to escape translational inhibition. While the precise molecular details of these observations remain to be elucidated, a short stem loop at the very 5′ end of the viral UTR, termed SL1, appears to play a critical role. SL1 is necessary but not sufficient to bypass the inhibition. Shi et al.[19] speculate that the study by Schubert et al.[17] had not detected this mechanism because the reporter constructs did not contain the short SL1 sequence—an attractive hypothesis that, however, will require dedicated further experiments for validation. In analogy to the SARS-CoV-1 findings, one may nevertheless speculate that the SL1–Nsp1 interaction would lead to the recruitment of host factors which enhance translation, and/or induce conformational changes within Nsp1 which induce its detachment from 40S.

**Host mRNA degradation through Nsp1.** In addition to the role in blocking translation, it is quite possible that SARS-CoV-2 Nsp1 also induces the degradation of host mRNA molecules. In so doing, the ratio of viral to host mRNA would be increased, and the production of viral proteins would be favoured. Of note, this hypothesis lacks direct evidence for the moment and is an extrapolation from findings in SARS-CoV-1 and MERS-CoV, where the corresponding Nsp1 orthologs possess such (endo) nucleolytic activity directed towards host mRNAs[9,12,22]. Cleaved host mRNAs lack their 5′ cap and are not only translationally inactive, but susceptible to full decay through the cellular degradation machinery. Notably, Lokugamage et al.[10] were able to identify a SARS-CoV-1 Nsp1 mutant protein (R124A, K125A) lacking mRNA cleavage activity. These amino acids are conserved in SARS-CoV-2 and the analogous mutant Nsp1 could represent an ideal starting point to explore whether a similar mRNA decay activity is associated with Nsp1 in this virus as well. For the moment, however, direct biochemical evidence of an intrinsic mRNA cleavage activity of SARS-CoV-2 Nsp1 is still lacking.

**Nsp1 autoregulation.** Even though the virus shifts translational capacity from host mRNA to its own RNA, a complete switch is (teleologically speaking) likely also not in the viral interest. In particular, it would be plausible that the virus has optimised the

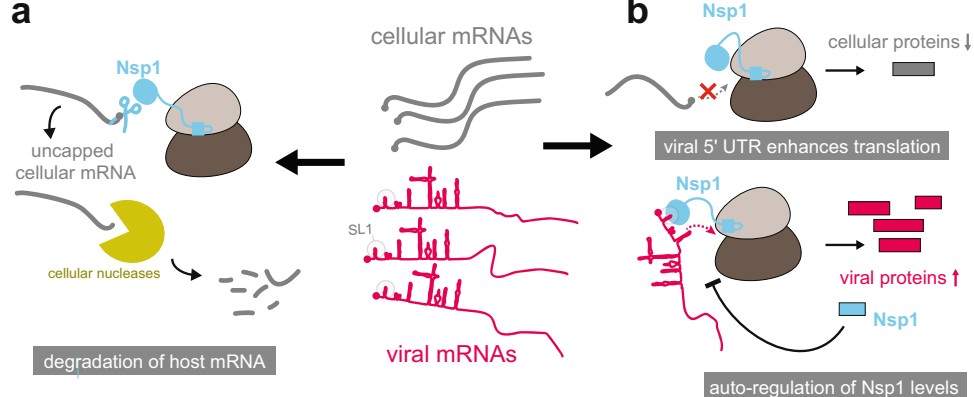

**Fig. 2 Nsp1 impacts host gene expression by several mechanisms.** Schematic representation of the main activities and mechanisms through which Nsp1 is thought to act in order to favour gene expression to viral transcripts, without shutting down mRNA translation completely. **a** Nsp1 may have a role in shifting the balance between viral and cellular RNAs in its favour, by inducing the cleavage/decapping of host mRNAs, which leads to their degradation by cellular nucleases. **b** The viral 5′ UTR (and in particular stem loop SL1) is likely a critical factor in directing ribosomes to the viral transcripts and overriding the translation block. Moreover, it has also been proposed that through Nsp1 autoregulation a total block of host mRNA translation may be prevented.

system in a way that host proteins necessary for viral replication can still be produced. First, one should consider that every mammalian cell harbours several million ribosomes[23]; it is unclear whether and with what kinetics during viral infection Nsp1 abundance can reach similar concentrations at all. Moreover, if viral mRNAs accumulate to very high levels—as suggested by the non-peer-reviewed study available as a preprint by Puray-Chavez et al.[24], who found that in Vero E6 cells more than 80% of RNA-seq reads were of viral origin 48 h post-infection—even relatively inefficient translation may be sufficient for viral reproduction. Finally, Schubert et al.[17] provide some evidence for Nsp1 autoregulation, which could contribute to establishing the optimal balance between a host translation inhibitory and permissive situation. Briefly, by transfecting equal amounts of Nsp1-encoding plasmid DNA into Hela cells, Schubert et al. observed a lower level of Nsp1 protein in cells transfected with wild-type Nsp1 than in cells transfected with Nsp1 that was mutated at its KH motif, potentially due to negative feedback of functional Nsp1 on its own translation. Further evidence will be required to understand the molecular basis and physiological relevance of the proposed negative feedback mechanism.

**Translational inhibition engenders a kinetic advantage over the host immune response**. A critical host response to viral infection is the activation of cell-intrinsic innate immune responses. RIG-I-like receptors (RLRs) are among the main actors in the detection of viral RNA and coronavirus infection[25]. Once activated, the RLR signalling cascade induces the expression of type I interferons (IFNs), which trigger innate antiviral immune responses aimed at suppressing viral replication and spreading at an early stage. These mechanisms are well established to occur in SARS-CoV-1 infection[14,26], yet SARS-CoV-2 may elicit them only poorly. Thoms et al.[18] investigated a potential involvement of Nsp1 in their suppression. They expressed a wild-type or mutant version (K164A, H165A; defective in 40S interaction) of Nsp1 in HEK293T cells and then activated the cellular RLR pathway. Wild-type Nsp1, but not the mutant protein, virtually shut down the translation of transcripts induced by IFN-β. Importantly, despite the strong reduction in translated proteins, the corresponding mRNA levels were not affected. It would therefore seem that the effect of Nsp1 is restricted to translational inhibition with little, if any, direct effect on immune response gene transcription and mRNA stability. It will be interesting to evaluate whether this effect on innate immune response gene expression is a reflection of the general block of translation or whether there is additional specificity for this class of transcripts. Finally, it will be important to evaluate to what extent we can extrapolate from such experiments in one specific, transformed cell line (HEK293T) that expresses Nsp1 in the absence of other coronaviral factors (but contains adenoviral E1a and E1b proteins), to a real SARS-CoV-2 infection. After all, the latter is associated with a robust, though delayed antiviral response, mediated by two RLRs, MDA5 and LGP2[27,28]. Nsp1 may be responsible for the observed delay, either through the translational inhibition it exerts or through other, additional mechanisms for which evidence is mounting. Several viral proteins (including Nsp1, when overexpressed) thus inhibit IFN induction by suppressing the activation of STAT1/2 transcription factors, which are critical effectors of the cascade[13,29–32]. Taken together, it is plausible that the multilevel interaction with the IFN response system will give SARS-CoV-2 a kinetic advantage over an immune response that is normally rapidly mounted. This characteristic appears to be one of the reasons why COVID-19 differs from SARS-CoV-1 and MERS infections[22], and Nsp1 seems to play a specific, critical role.

**Nsp1—Achilles' heel of SARS-CoV-2?** Given the important functions of Nsp1 that have been revealed, could this protein actually constitute a vulnerability of the virus relevant for the development of a future drug or vaccine? Conceptually, a drug designed to target Nsp1 would need to prevent its binding to the ribosome without interfering with ribosomal function, thus allowing the cellular defence systems to mount a response. As recently shown by Xia et al.[29], the development of small molecule drugs targeting ribosomal RNAs could be a possible strategy to disrupt the interaction between Nsp1 and 18S rRNA. Another promising strategy could lie in targeting the 5′ viral leader; indeed, if the first loop of the stem (SL1) is sufficient to prevent the suppression of translation during the expression of Nsp1, as suggested by Banerjee et al.[20], it might be possible to design small molecules or antisense oligonucleotides that bind specifically to the relevant part of the RNA.

From a public health perspective, the most important approach to combat the devastating infectious impact of SARS-CoV-2 lies in the development of vaccines. We are seeing significant progress at the moment in this regard, with several efficient vaccines on the market. Nevertheless, given that vaccination will need to stop the replication of the virus globally and that vaccine escaper variants will likely emerge over time, it will remain of importance to develop additional vaccines and treatments to cure infected individuals as well. Vaccines are typically designed using proteins that are on the surface of the viral particles. Nevertheless, if Nsp1 is as essential as suggested for an infection and likely does not easily tolerate mutations that would help evade immune system recognition, the design of a vaccine based on this protein could be an interesting complementary strategy. Also, an attenuated virus, e.g. lacking the essential KH motif that is critical for ribosome binding and translation inhibition, could potentially be envisioned as it would enable an effective host immune response in addition to generating the immune memory essential to combat new SARS-CoV-2 infections.

## Conclusion

Historically, many important discoveries in molecular biology have been made through the study of viruses. The fascinating structural work on Nsp1–ribosome complexes is enlightening for our fundamental understanding of cellular processes and their hijacking during viral attack. Although SARS-CoV-2 is becoming better understood day by day, much research is still needed, in particular to understand how the various effects discussed above in isolation are integrated together (e.g. those on translation, with those presumably acting on host mRNA abundance), thus leading to the reprogramming of the host cell gene expression landscape. Surprising (and sometimes contradictory) findings show us that we are far from fully understanding the complexity of the system. For instance, a recent study by Finkel et al.[33] has shown that viral mRNAs are not translated more efficiently than host mRNAs, in apparent contradiction to some of the data discussed above. Instead, the authors propose that it is simply the high levels of viral transcripts that explain how viral translation dominates host translation. Also, a detailed time-course study of the transcriptome of SARS-CoV-2-infected cells would be revealing to identify which host genes are directly impacted by the virus. A first step in this direction is reported in a non-peer-reviewed study that is available as a preprint, by Puray-Chavez et al.[24], who use ribosome profiling in SARS-CoV-2-infected cells to follow temporal changes at the viral and host RNA and translation level, allowing insights into how translational regulation impacts SARS-CoV-2 replication and host cell survival. Undoubtedly, many additional, complementary studies will appear in the near future. They will help us to understand the biology of SARS-CoV-2,

which is directly relevant to medical progress that is needed to combat the current pandemic and to prepare for future pandemics. Finally, given the wealth of high-quality fundamental research on a virus that was first described only some months ago, one of the most important take-home messages may be that modern science can progress at an extraordinary pace, especially when the scientific community pulls together.

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

## Acknowledgements

Research in the laboratory of D.G. was funded by the University of Lausanne and by the Swiss National Science Foundation through the National Center of Competence in Research RNA & Disease (grant no. 141735) and individual grant 179190.

## Author contributions

The first draft was written by M.S., T.C. and D.R. ("Write a Review" course within the master of science in Molecular Life Sciences program at the University of Lausanne, supervision: D.G.). D.G. further adapted the manuscript text and the figures. All authors reviewed the final manuscript.

## Competing interests

The authors declare no competing interests.
