## [Transparent Peer Review File · Communications Biology]

Reviewers' comments:

Reviewer #1 (Remarks to the Author):

This review focuses on recent advancements that have been made to understand the role of SARS-CoV2 coronavirus nonstructural protein 1 (nsp1) in suppressing translation by the host ribosome. The review also looks at mechanisms that sustain translation of viral proteins despite global nsp1-mediated translational suppression. Future therapies and vaccine possibilities targeting nsp1 are also discussed.

Overall, the manuscript is well written, highly interesting and timely. It brings together research on an important mechanism used by SARS-CoV-2 to hijack the host ribosome. The review is quickly digestible and informative. Given the volume of research emerging on this virus, focused reviews like this are welcome. I only have very minor comments.

Comments:

Line 55: This sentence is awkward: "We also discuss by which mechanisms sufficient viral translation could be maintained despite global, Nsp1-mediated translational repression."

Line 84: Change "consisted in" to "consisting of"

Line 113: Change "adapts" to "adopts"

Line 185: Describe the Nsp1 mutant used by Thoms et al in this experiment and the rationale.

Reviewer #2 (Remarks to the Author):

This is a very timely review and the authors did a great job in comparing several studies. I think a few more studies should be cited and these are highlighted below.

1- As far as RLR involvement in SARS-CoV-2 infection, I would add two recent published studies that demonstrate MDA5-mediated sensing of SARS CoV-2 culminating in seemingly robust type I IFN immunity (one by Sumit Chanda's lab and the other one by Caroline Goujon). There is some evidence that this response is a bit "delayed", so one can possibly argue that this delay may be mediated by Nsp1. Obviously without proper reverse genetics approaches to generate Nsp1 deleted virus this is very difficult to test in infected cells.

2- Lines 196-199: There are now also numerous other studies that suggest the involvement of other viral proteins in blocking IFN induction through antagonism of STAT1/2. In addition to nsp1, these include nsp6, nsp13, ORF3a, M, ORF7a and ORF7b which can inhibit STAT1/2 phosphorylation and ORF6 which can inhibit STAT1 nuclear translocation. See references [1-5] below.

1. Sa Ribero, M., et al., Interplay between SARS-CoV-2 and the type I interferon response. *PLoS Pathog*, 2020. 16(7): p. e1008737.
2. Lei, X., et al., Activation and evasion of type I interferon responses by SARS-CoV-2. *Nat Commun*, 2020. 11(1): p. 3810.
3. Konno, Y., et al., SARS-CoV-2 ORF3b Is a Potent Interferon Antagonist Whose Activity Is Increased by a Naturally Occurring Elongation Variant. *Cell Rep*, 2020. 32(12): p. 108185.
4. Miorin, L., et al., SARS-CoV-2 Orf6 hijacks Nup98 to block STAT nuclear import and antagonize interferon signaling. *Proc Natl Acad Sci U S A*, 2020. 117(45): p. 28344-28354.
5. Xia, H., et al., Evasion of Type I Interferon by SARS-CoV-2. *Cell Rep*, 2020. 33(1): p. 108234.

3- I would also discuss whether the physiologically relevant concentrations of Nsp1 will be able to induce translational block in infected cells given that ribosomes are so abundant (and I greatly doubt nsp1 will be as abundant). The Puray-Chavez and Finkel studies both indicate that viral RNAs are so abundant in the cells that they do not have to be translated efficiently. In fact Puray-Chavez study shows that viral RNAs constitute up to 80% of mRNAs in infected cells.

Reviewer #3 (Remarks to the Author):

Simeoni et al present a timely, balanced review of the multiple activities encoded by coronavirus nsp1 proteins and the exciting, recently solved cryo-EM structures of SARS-CoV-2 nsp1 in association with 40S subunits and 80S ribosomes. They also present a provocative preview of data in a bioRxiv preprint that suggests a stem loop structure contained within SARS-CoV2 5'UTR supports viral mRNA translation by overcoming nsp1-mediated inhibition. The review is well written and will be of general interest to researchers investigating coronavirus infection biology, post-transcriptional control of gene expression, and virus-host interactions that influence host innate defenses.

My suggestions to improve the review are below:

Line 43: instead of "immune system", it would be more informative to refer to cell intrinsic, innate immune responses.

Line 43-44: "Of note, the mechanisms by which Nsp1 proteins operate may vary for different beta-CoVs,....." is in need of specific references / citations.

Lines 50-52 is redundant with earlier statements (line 43,44)

Line 63: Revise to read "For SARS-CoV-1..."

Line 63-67: Has the extent to which SARS-CoV-2 nsp1 inhibits translation been determined experimentally and compared to CoV-1 nsp1? This should be addressed directly (in addition to the conserved residues).

Lines 94-96: It should be mentioned that the physiological significance of these exotic complexes remains unknown, especially within cells infected with SARS-CoV-2.

Line 118: Do the authors mean to cite ref 17 with respect to nsp1 impacting immune functions?

Line 136-144: While the claims in the preprint ref 18 are indeed exciting, a more critical approach to the story might be helpful. Further genetic analysis of the SL1 sequence is required to establish the role of sequence vs base pairing structure (ideally mutants that disrupt base pairing compared to compensatory mutations that restore structure but not sequence). And more work is needed to determine whether differences in findings between ref 15 vs 18 do indeed result from differences surrounding the cis-elements in the reporter constructs (or not).

Line 144: change "comprise" to "contain".

Line 145-146: the interaction between a 5' UTR-borne SL1 and Nsp1 that has been reported to play a role in translational shut-off evasion in SARS-CoV-1 should be presented prior to the Shi et al preprint. This work should be presented first given that it was performed in 2012 and likely established the premise for the SARS-CoV-2 findings.

Line 156-158: please specify / identify precisely what kind of RNA chemical modifications are found near the cap site

Line 162: The relevance of this statement is questioned: " Uniprot [21], for example, already infers this activity from sequence similarity." ^[1]_{SEP}

Lines 161-166: The authors should point out that direct biochemical evidence that the SARS-CoV-2 nsp1 contains an intrinsic mRNA cleaving activity is lacking and needs to be established.

Line 173-176: The evidence in support of nsp1 negative feedback on its translation seems flimsy based upon the data described and the questionable physiological relevance of the assay system.

Lines 182-190: The authors should make clear that conclusions reached regarding Wild-type Nsp1, but not the mutant protein upon transient expression were derived from experiments performed in a non-physiological, engineered established, transformed cell line (293 cells, which express Adenovirus E1a and E1b proteins. Also, E1b influences mRNA post-transcriptional processes and could in unknown ways skew the results). This does not necessarily mean they are incorrect, but different findings regarding roles of RNA decay vs translational control might emerge in a more physiologically based experimental system. ^[1]_{SEP}

Lines 193-195: this sentence ".....the virus will start doing mischief with a kinetic advantage over an immune response that is normally rapidly mounted. " requires correction and rewriting. Viruses do not do mischief.

Lines 196-198: were the studies in question overexpression studies of individual proteins? This should be noted.

Line 218: "so presumably this ship has sailed". I think this is a statement is not warranted. To effectively stop the generation of variants, virus replication must be stopped globally through vaccination campaigns. Prior experience has shown just how difficult a global eradication plan such as this will be. There very well may be a continued need to develop / modify vaccines as variants emerge in populations that are not sufficiently vaccinated and to develop therapeutics to treat infected individuals.

What about the need for therapeutics active against emerging variants — in the interval(s) where variants may not be covered by vaccines?

Line 241-243: does the quality / strength of the data in ref 27 preprint support the claim that structural elements regulate viral gene expression?

We thank the reviewers for their constructive criticism and suggestions on how to improve the manuscript. We have addressed all issues, as described below.

Reviewer #1 (Remarks to the Author):

This review focuses on recent advancements that have been made to understand the role of SARS-CoV2 coronavirus nonstructural protein 1 (nsp1) in suppressing translation by the host ribosome. The review also looks at mechanisms that sustain translation of viral proteins despite global nsp1-mediated translational suppression. Future therapies and vaccine possibilities targeting nsp1 are also discussed.

Overall, the manuscript is well written, highly interesting and timely. It brings together research on an important mechanism used by SARS-CoV-2 to hijack the host ribosome. The review is quickly digestible and informative. Given the volume of research emerging on this virus, focused reviews like this are welcome. I only have very minor comments.

Author Response: We thank the Reviewer for the positive evaluation. We have addressed the comments as described in the following.

Comments:

Line 55: This sentence is awkward: “We also discuss by which mechanisms sufficient viral translation could be maintained despite global, Nsp1-mediated translational repression.”

Author Response: We have changed the sentence to render it less awkward.

Line 84: Change “consisted in” to “consisting of”

Author Response: The sentence has been re-written.

Line 113: Change “adapts” to “adopts”

Author Response: The sentence has been changed.

Line 185: Describe the Nsp1 mutant used by Thoms et al in this experiment and the rationale.

Author Response: We have added this information.

Reviewer #2 (Remarks to the Author):

This is a very timely review and the authors did a great job in comparing several studies. I think a few more studies should be cited and these are highlighted below.

Author Response: We thank the Reviewer for the positive evaluation. We have addressed the comments as described in the following.

1- As far as RLR involvement in SARS-CoV-2 infection, I would add two recent published studies that demonstrate MDA5-mediated sensing of SARS CoV-2 culminating in seemingly robust type I IFN immunity (one by Sumit Chanda's lab and the other one by Caroline Goujon). There is some evidence that this response is a bit "delayed", so one can possibly argue that this delay may be mediated by Nsp1. Obviously without proper reverse genetics approaches to generate Nsp1 deleted virus this is very difficult to test in infected cells.

Author Response: We now cite this data and both papers, and explain the idea of Nsp1 "delaying" the IFN response.

2- Lines 196-199: There are now also numerous other studies that suggest the involvement of other viral proteins in blocking IFN induction through antagonism of STAT1/2. In addition to nsp1, these include nsp6, nsp13, ORF3a, M, ORF7a and ORF7b which can inhibit STAT1/2 phosphorylation and ORF6 which can inhibit STAT1 nuclear translocation. See references [1-5] below.

1. Sa Ribero, M., et al., Interplay between SARS-CoV-2 and the type I interferon response. PLoS Pathog, 2020. 16(7): p. e1008737.

2. Lei, X., et al., Activation and evasion of type I interferon responses by SARS-CoV-2. Nat Commun, 2020. 11(1): p. 3810.

3. Konno, Y., et al., SARS-CoV-2 ORF3b Is a Potent Interferon Antagonist Whose Activity Is Increased by a Naturally Occurring Elongation Variant. Cell Rep, 2020. 32(12): p. 108185.

4. Miorin, L., et al., SARS-CoV-2 Orf6 hijacks Nup98 to block STAT nuclear import and antagonize interferon signaling. Proc Natl Acad Sci U S A, 2020. 117(45): p. 28344-28354.

5. Xia, H., et al., Evasion of Type I Interferon by SARS-CoV-2. Cell Rep, 2020. 33(1): p. 108234.

Author Response: We now cite all five suggested references in the corresponding part of the manuscript (in the first version, only the last publication by Xia et al. had been included). We refrained from precisely laying out the various findings of these studies, as we felt that they would add unnecessary detail to the manuscript.

3- I would also discuss whether the physiologically relevant concentrations of Nsp1 will be able to induce translational block in infected cells given that ribosomes are so abundant (and I greatly doubt nsp1 will be as abundant). The Puray-Chavez and Finkel studies both indicate that viral RNAs are so abundant in the cells that they do not have to be translated efficiently.

In fact Puray-Chavez study shows that viral RNAs constitute up to 80% of mRNAs in infected cells.

Author Response:

These are excellent suggestions that we have now included in the paragraph "*Nsp1 autoregulation.*"

Reviewer #3 (Remarks to the Author):

Simeoni et al present a timely, balanced review of the multiple activities encoded by coronavirus nsp1 proteins and the exciting, recently solved cryo-EM structures of SARS-CoV-2 nsp1 in association with 40S subunits and 80S ribosomes. They also present a provocative preview of data in a bioRxiv preprint that suggests a stem loop structure contained within SARS-CoV2 5'UTR supports viral mRNA translation by overcoming nsp1-mediated inhibition. The review is well written and will be of general interest to researchers investigating coronavirus infection biology, post-transcriptional control of gene expression, and virus-host interactions that influence host innate defenses.

Author Response: We thank the Reviewer for the positive evaluation. We have addressed the comments as described in the following.

My suggestions to improve the review are below:

Line 43: instead of "immune system", it would be more informative to refer to cell intrinsic, innate immune responses.

Author Response: We have made the suggested change.

Line 43-44: "Of note, the mechanisms by which Nsp1 proteins operate may vary for different beta-CoVs,....." is in need of specific references / citations.

Author Response: We have updated this part of the manuscript, including through specific references.

Lines 50-52 is redundant with earlier statements (line 43,44)

Author Response: We have removed the redundancy.

Line 63: Revise to read "For SARS-CoV-1..."

Author Response: The sentence has been changed.

Line 63-67: Has the extent to which SARS-CoV-2 nsp1 inhibits translation been determined experimentally and compared to CoV-1 nsp1? This should be addressed directly (in addition to the conserved residues).

Author Response: To the best of our knowledge, no such data is currently available.

Lines 94-96: It should be mentioned that the physiological significance of these exotic complexes remains unknown, especially within cells infected with SARS-CoV-2.

Author Response: We have made it clearer that the physiological significance of these exotic complexes remains unknown.

Line 118: Do the authors mean to cite ref 17 with respect to nsp1 impacting immune functions?

Author Response: This was a miscitation from our side that we have now corrected.

Line 136-144: While the claims in the preprint ref 18 are indeed exciting, a more critical approach to the story might be helpful. Further genetic analysis of the SL1 sequence is required to establish the role of sequence vs base pairing structure (ideally mutants that disrupt base pairing compared to compensatory mutations that restore structure but not sequence). And more work is need to determine whether differences in findings between ref 15 vs 18 do indeed result from differences surrounding the cis-elements in the reporter constructs (or not).

Author Response: We now clearly indicate the speculative nature of the hypotheses surrounding SL1.

Line 144: change “comprise” to “contain”.

Author Response: We have changed “comprise” to “contain”.

Line 145-146: the interaction between a 5' UTR-borne SL1 and Nsp1 that has been reported to play a role in translational shut-off evasion in SARS-CoV-1 should be presented prior to the Shi et al preprint. This work should be presented first given that it was performed in 2012 and likely established the premise for the SARs-CoV-2 findings.

Author Response: We have changed the order as suggested.

Line 156-158: please specify / identify precisely what kind of RNA chemical modifications are found near the cap site.

Author Response: We have changed these sentences and now only speak of the (endo)nucleolytic cleavage.

Line 162: The relevance of this statement is questioned: " Uniprot [21], for example, already infers this activity from sequence similarity."

Author Response: We have removed this statement.

Lines 161-166: The authors should point out that direct biochemical evidence that the SARS-CoV-2 nsp1 contains an intrinsic mRNA cleaving activity is lacking and needs to be established.

Author Response: We have made this clearer, as suggested.

Line 173-176: The evidence in support of nsp1 negative feedback on its translation seems flimsy based upon the data described and the questionable physiological relevance of the assay system.

Author Response: We have toned down the description and interpretation of this data.

Lines 182-190: The authors should make clear that conclusions reached regarding Wild-type Nsp1, but not the mutant protein upon transient expression were derived from experiments performed in a non-physiological, engineered established, transformed cell line (293 cells, which express Adenovirus E1a and E1b proteins. Also, E1b influences mRNA post-transcriptional processes and could in unknown ways skew the results). This does not necessarily mean they are incorrect, but different findings regarding roles of RNA decay vs translational control might emerge in a more physiologically based experimental system.

Author Response: We have added this caveat, as suggested.

Lines 193-195: this sentence ".....the virus will start doing mischief with a kinetic advantage over an immune response that is normally rapidly mounted. " requires correction and rewriting. Viruses do not do mischief.

Author Response: The sentence has been re-written.

Lines 196-198: were the studies in question overexpression studies of individual proteins? This should be noted.

Author Response: This indeed referred to data in which individual tagged viral proteins were overexpressed. We have now made this clearer.

Line 218: "so presumably this ship has sailed". I think this is a statement is not warranted. To

effectively stop the generation of variants, virus replication must be stopped globally through vaccination campaigns. Prior experience has shown just how difficult a global eradication plan such as this will be. There very well may be a continued need to develop / modify vaccines as variants emerge in populations that are not sufficiently vaccinated and to develop therapeutics to treat infected individuals.

What about the need for therapeutics active against emerging variants — in the interval(s) where variants may not be covered by vaccines?

Author Response: We thank the Reviewer for these valuable comments. This part of the manuscript has been updated accordingly.

Line 241-243: does the quality / strength of the data in ref 27 preprint support the claim that structural elements regulate viral gene expression?

Author Response: These sentences have been rewritten and toned down, in line with the fact that the study is still at the preprint stage.

REVIEWERS' COMMENTS:

Reviewer #3 (Remarks to the Author):

The authors have nicely addressed all my comments.